# Sex differences in sexual attraction for aesthetics, resources and personality across age

Stephen Whyte[1,2]*, Robert C. Brooks[3], Ho Fai Chan[1], Benno Torgler[1,4]

**1** School of Economics and Finance & Centre for Behavioural Economics, Society & Technology (BEST), Queensland University of Technology, Gardens Point, Brisbane, QLD, Australia, **2** Centre in Regenerative Medicine, Institute of Health and Biomedical Innovation, Queensland University of Technology (QUT), Kelvin Grove, QLD, Australia, **3** Evolution & Ecology Research Centre, School of Biological, Earth and Environmental Sciences, University of New South Wales, Sydney, NSW, Australia, **4** CREMA—Center for Research in Economics, Management and the Arts, Basel, Switzerland

⊙ These authors contributed equally to this work.
* sg.whyte@qut.edu.au.

**Data Availability Statement:** Data and codes used in this study can be found on the Open Science Framework (DOI: 10.17605/OSF.IO/DSJ9W).

**Funding:** The author(s) received no specific funding for this work.

## Abstract

Because sexual attraction is a key driver of human mate choice and reproduction, we descriptively assess relative sex differences in the level of attraction individuals expect in the aesthetic, resource, and personality characteristics of potential mates. As a novelty we explore how male and female sexual attractiveness preference changes across age, using a dataset comprising online survey data for over 7,000 respondents across a broad age distribution of individuals between 18 and 65 years. In general, we find that both males and females show similar distribution patterns in their preference responses, with statistically significant sex differences within most of the traits. On average, females rate age, education, intelligence, income, trust, and emotional connection around 9 to 14 points higher than males on our 0–100 scale range. Our relative importance analysis shows greater male priority for *attractiveness* and *physical build*, compared to females, relative to all other traits. Using multiple regression analysis, we find a consistent statistical sex difference (males relative to females) that decreases linearly with age for *aesthetics*, while the opposite is true for *resources* and *personality*, with females exhibiting a stronger relative preference, particularly in the younger aged cohort. Exploring non-linearity in sex difference with contour plots for intelligence and attractiveness across age (mediated by age) indicates that sex differences in attractiveness preferences are driven by the male cohort (particularly age 30 to 40) for those who care about the importance of age, while intelligence is driven by females caring relatively more about intelligence for those who see age as very important (age cohort 40 to 55). Overall, many of our results indicate distinct variations within sex at key life stages, which is consistent with theories of selection pressure. Moreover, results also align with theories of parental investment, the gender similarities hypothesis, and mutual mate choice– which speaks to the fact that the broader discipline of evolutionary mate choice research in humans still contains considerable scope for further inquiry towards a unified theory, particularly when exploring sex-difference across age.

**Competing interests:** The authors have declared that no competing interests exist.

## Introduction

Sexual attraction is a primary driver of mate selection and reproductive decisions, and those decisions impact not only the individuals involved but the broader market in which the decision occurs [1]. Micro level decision making on sex, reproduction, and relationship formation thus influences a wide variety of macro trends and social norms, including, *inter alia*, gender roles and equity, labor market dynamics, fertility rates, wider sexual liberalism, religion, politics, and the broader institution of marriage [2–4]. However, there is ongoing debate inside the field of evolutionary mate choice regarding exactly how similar or different male and female preferences are across age [5–8]. To investigate competing evolutionary theories, we analyze responses from more than 7000 Australian online dating participants to generate a descriptive overview of both *relative* and *absolute* sex differences in individuals' stated preferences for *aesthetics*, *resources*, and *personality* characteristics in a potential mate across age.

As argued in numerous scientific disciplines, the human preference for attractive mates and the ability to quickly identify such attractiveness in others [9] reflect an evolved adaptive preference to reproduce "good genes" [10] And while both sexes prefer mates who are physically attractive and state clear preferences for the level of attractiveness sought in a potential partner, males have been shown to report stronger preferences for attractiveness relative to females [11]. Trivers' 1972 [12] seminal work argued that these different preferences between the sexes for particular characteristics in a mate should clearly reflect sexual selection pressures. That is, females are more selective, not only because their maximum fecundity is time limited but because choosing poorly increases the long-term opportunity costs of reproduction (internal gestation, ongoing lactation, and disproportionate maternal investment) and reduces the probability of offspring [12]. Their mate preferences should thus reflect characteristics or traits that can compensate for disproportionate maternal investment and ensure offspring survival and reproductive success, especially with respect to economic proxies for resources and/or increased paternal investment such as educational level, intelligence, and income. In fact, research has shown females demonstrate far more stringent preferences than males for mates with good earning potential or higher education [11], particularly during the years of peak fertility [13, 14]. Males, in contrast, need only invest the time taken to copulate, which paucity of paternal investment implies the favoring of mates whose genetic fitness guarantees a maximum chance of offspring survival and reproduction.

Accordingly, based on an assumption that aesthetics such as age, attractiveness, and symmetry of physical build or features imply a lower likelihood of disease or pathogen prone ancestry, humans use these instantly recognizable characteristics as proxies for both genetic and phenotypic condition. Not only are males more likely than females to state a preference for physically attractive characteristics in a mate [15], but their prioritizing of female facial cues over body shape is dependent on the planned mating duration [16]. That is, whereas females remain unaffected by mating temporality, males prioritize facial cues in a long-term mating context but bodily cues in a short-term one. This difference may stem from female faces and bodies simultaneously showcasing traits that are pronounced correlates of health and fertility, such as estrogen-dependent facial features (lips, cheeks, jaw line) and body features (waist-to-hip ratio and accentuated gait) p. 490 [17].

What attracts an individual to a mate, therefore–what the average human finds "sexy"–is an integral part of short- or long-term mate choice strategies and inherently a key point of similarity or difference between the sexes. More recently, a new body of literature has begun to emerge that is more critical of the good genes hypothesis [18–20], and psychology's possible overstatement of sex differences in human mate choice [6]. While there is indeed a large sex difference in obligate parental investment in humans, sex differences in "typical" parental

investment (particularly in modern developed economies) are much smaller, which ultimately leads to similar levels of choosiness in long-term mating domains. This new body of literature points out that, even cross-culturally, certain favored mate choice traits or characteristics are still important for both sexes [21] and that the dynamics of mutual mate choice (MMC) reveal sex differences that are more appropriately characterized as relative, rather than absolute.

This study therefore descriptively examines the responses of a large sample of Australian online dating participants ($n$ = 7325) to assess level of sexual attraction in a mate expressed as stated preferences by both sexes. Our key focus is an exploration of how relative sex preferences change across age as limited empirical evidence and theoretical understandings are available regarding such potential preference changes. Most studies on sexual attractiveness rely on a limited age distribution skewed towards the younger population. We therefore utilize a large age distribution (18–65 years of age) of a cross-sectional data set to better understand relative sex differences across age for different individuals' sexual attraction towards *aesthetic*, *resource*, and *personality* traits in a potential mate. Thus, our main goal is empirical rather than theoretical, as the literature behind sex preference differences makes no or very limited predictions for how this pattern might change with age.

## Aesthetics, resources and personality in mate choice

### Aesthetics

Attractive individuals derive a broad social range of utility, enjoying everything from greater choice in the mating market to greater human capital investment during one's education, and even increased returns from their labor market productivity [22–24]. This may be because judgements on attractiveness potentially reflect evaluation of apparent physical health [25] or phenotypic condition. In fact, applying the biological "good genes" hypothesis specifically to sexual selection implies that the mate preference for healthy looking (i.e., attractive) individuals who promise the associated direct and indirect benefits is an adaptive preference for physical traits that increase both parental and offspring fitness [10]. Of course, it is reasonable to expect differences in the use of aesthetic indicators of fertility and reproductive capacity such as age, attractiveness, and physical build and features. For example, whereas the human female has a relatively short reproductive phase and declining fecundity with age [26], human males often maintain their reproductive function (with only minimal decline) until old age [27]. And while older males are more prone to the rare spontaneous de novo mutations that can increase the risk of conditions like autism, there is no critical threshold for sperm production, and men can realize offspring far beyond their 40s [28]. Therefore, evolutionary science has theorized and demonstrated that males are more likely to state a preference for females at peak fertility [11] even when they themselves are beyond this stage [29], whereas females are more likely to refine their specific mate preferences across their years of peak fertility [11]. Yet, the size of the relative difference in both sexes' preference for aesthetics in a mate–at different life stages– remains unclear.

### Resources

Because of the human female's disproportionate opportunity costs of gestation and lactation, males have historically been rewarded (or at least not disadvantaged like females) in both higher education and the labor market for their ability to work continuously without reproduction-related career interruptions [2]. The fact that this freedom to continue attracting income translates into increased human capital, higher wage rates, and greater lifetime earnings may partly explain some of the current and historical large intersex variance in the ability to access, possess, and accrue resources [11]. For example, according to the Australian Bureau

of Statistics [30], average hourly earnings peak for males in the mid-40s but for females, earnings peak in their 30s [31]. Such wage gaps are common and consistent across developed economies [32], accentuating male access to earnings and resources, and forcing women who seek higher education or equivalent earnings in their late 20s through their 30s to postpone or delay pregnancy [27]. As mate choice research across a myriad of disciplines repeatedly demonstrates, this combination of disproportionate physiological investment and constraints in accessing resources places evolutionary selection pressure on females to secure or favor mates who can compensate for this cost constraint [1, 11, 14, 33]. With such disproportionate opportunity costs for both labor market and reproductive participation, sex differentiated preferences for accrued resources or proxies for access should also be visible in any mating market. Furthermore, because income is positively correlated with age (on average), but shows diminishing returns beyond 50 years of age [31], relative and absolute preference for resources potentially also change as a function of age for both sexes.

## Personality

Given the inheritability of personality traits [34], if males exhibit particular personality traits that signal (increased) paternal investment, it seems likely that females have also evolved specialized mechanisms that suggest corresponding maternal characteristics. In addition to increasing pair bond strength through parental investment, such positive externalities in mate choice may also reinforce reciprocally altruistic behavior between mates [12], increase complementary production in the household [2], promote kin selection towards genetic relatives [3], and increase the chances of long-term mate retention. Their importance in mate selection may also be increasing in developed countries where sex-discrimination legislation and wider efforts towards gender equity have narrowed the gap in the ability of males and females to acquire the income and wealth resources that benefit child rearing and welfare. Hence, modern female preferences may not only more acutely favor personality traits or "good father attributes" that increase reproductive success [35], but as household, gender, and labor market roles evolve and even converge, personality traits may become a greater point of differentiation [36] in a potential mate than resources or the ability to acquire them. The current study provides a unique opportunity to explore both relative and absolute sex difference stated preference for key personality factors such as trust, openness, and emotional connection in a large sample ($n = 7325$) of online dating participants and how those preferences change with age. The way in which sex preference differences for personality change with age is theoretically as well as empirically underexplored. Thus, due to the limited existing theoretical understanding, we try to contribute to the area by providing novel empirical insights that may guide future theoretical insights–as science can be seen as a constant interaction between speaking to theorists and searching for facts.

## Method

### Research design

Our analysis is based on participant responses to nine different versions of the same question format, covering nine characteristics associated with sexual attraction–age, attractiveness, physical build/features, intelligence, education, income, trust, openness, emotional connection:

To what extent do you find a person's [SPECIFIC TRAIT] influences how sexually attractive you find them:

0 = Not important at all . . . . . . . . . . . . . . . . . . . . . . . . . . . .... . .. Extremely important = 100.

Each question thus asks respondents to rate the level of importance they assign to each characteristic in relation to sexual attraction on a sliding scale from 0 to 100. These characteristics are then grouped into three key categories commonly associated with sexual attraction: aesthetics (age, attractiveness, and physical build/features), resources (intelligence, education, and income), and personality (trust, openness, and emotional connection). The aesthetic factors are easily recognizable and assessable in even minute interpersonal interactions or exposure [37]; the three resource factors are all commonly used in mate choice research, as they aid parental investment [1, 11, 38]; and finally, the three personality factors matter for interpersonal relationships, pair bonding, and parental investment [39–41].

## Data collection

These data were collected as part of the national online Australian Sex Survey, administered to the Australian general public between July 25 and September 19, 2016, and resulting in a very broad Australian sample. Some data from the survey has already been published in unrelated research [36, 42–45]. Participation was incentivized by three random draws for approximately $1,500 worth of prizes donated by the industry partners Adultmatchmaker.com and its affiliated dating web sites, Eros Association, the Australian Sex Party, Max Black, and Giga Pty Ltd. All research was conducted in accordance with Queensland University of Technology (QUT) human research ethics on clearance approval number 1600000221. All participants were 18 years of age or older at the time of the survey, and provided written informed consent to participate (see A1 Table in S1 Appendix for the summary statistics of the sample, by sex).

## Results

### Descriptive results

We first explore the distribution of the responses to the nine characteristics, differentiated by sex (see A1 Fig in S1 Appendix). Overall, the distribution between the nine traits follows a similar pattern for both sexes; for example, the three personality traits, physical build, and attractiveness, are rated quite high on the importance scale for both sexes, while age, intelligence, and education are more evenly rated, and income is rated quite low on the importance scale. Table 1 shows significant within-trait sex differences for 8 out of 9 traits. In particular, we find that, on average, females rate age (Cliff's delta $\delta = 0.255$, $p<0.001$), education ($\delta = 0.253$, $p<0.001$), intelligence ($\delta = 0.309$, $p<0.001$), income ($\delta = 0.25$, $p<0.001$), trust ($\delta = 0.222$, $p<0.001$), and emotional connection ($\delta = 0.309$, $p<0.001$) between 9 and 14 points higher than males do (on a scale ranging from 0 to 100). On the other hand, there is no statistical sex difference in terms of importance rating in the attractiveness attribute ($\delta = 0.013$, $p = 0.730$), and the difference in physical build ($\delta = 0.039$, $p = 0.121$) is minimal. One should also note that the overlapping coefficients for the (male and female) distributions for attractiveness, physical build, and openness are among the highest. The overlapping coefficient indicates the degree of overlap between the kernel density estimates of the respective distribution (male and female). For example, a value of 1 would indicate a perfect overlap between the two distributions [46, 47].

To gauge the relative importance of the nine characteristics for each individual participant, we standardized the responses to the nine traits 'within' subject. Specifically, for each individual, we first calculate the average value (level) of the nine responses as well as the standard deviation (spread), then we subtract the value of each trait to this average and divide by the standard deviation. Essentially, one can interpret the standardized values as the importance of one trait relative to the average importance of all nine traits. This approach safeguards the results from comparison with omitted factors (e.g., common interests), which might cause

**Table 1. Sex preference differences in characteristics.**

| Absolute importance | Mean Diff. | Cliff's delta | z-stat. (two tailed) | Males | | | | Females | | | | Overlap. Coeff. |
|---|---|---|---|---|---|---|---|---|---|---|---|---|
| | | | | Mean | SE | 95% CI Lower | 95% CI Upper | Mean | SE | 95%CI Lower | 95%CI Upper | |
| Aesthetics | | | | | | | | | | | | |
| Age | 11.927 | 0.255 | 18.01*** | 48.1 | 0.41 | 47.3 | 48.9 | 60.0 | 0.48 | 59.1 | 61.0 | 0.700 |
| Attractiveness | -0.187 | 0.013 | -0.89 | 65.9 | 0.34 | 65.3 | 66.6 | 65.7 | 0.42 | 64.9 | 66.6 | 0.928 |
| Physical build | -0.838 | 0.039 | -2.77* | 65.3 | 0.35 | 64.7 | 66.0 | 64.5 | 0.42 | 63.7 | 65.3 | 0.898 |
| Resources | | | | | | | | | | | | |
| Education | 12.172 | 0.253 | 17.88*** | 41.2 | 0.41 | 40.4 | 42.1 | 53.4 | 0.52 | 52.4 | 54.4 | 0.689 |
| Intelligence | 13.894 | 0.309 | 21.86*** | 55.8 | 0.40 | 55.0 | 56.6 | 69.7 | 0.43 | 68.9 | 70.6 | 0.640 |
| Income | 9.695 | 0.25 | 17.69*** | 19.6 | 0.31 | 19.0 | 20.2 | 29.3 | 0.45 | 28.4 | 30.2 | 0.675 |
| Personality | | | | | | | | | | | | |
| Openness | 4.445 | 0.132 | 9.35*** | 69.2 | 0.31 | 68.6 | 69.9 | 73.7 | 0.37 | 73.0 | 74.4 | 0.817 |
| Trust | 9.143 | 0.222 | 15.66*** | 68.9 | 0.38 | 68.2 | 69.7 | 78.1 | 0.40 | 77.3 | 78.9 | 0.717 |
| Emotional connection | 12.397 | 0.309 | 21.86*** | 65.1 | 0.39 | 64.3 | 65.9 | 77.5 | 0.42 | 76.7 | 78.3 | 0.623 |

| Relative importance | Mean Diff. | Cohen's d | t-stat. (two tailed) | Males | | | | Females | | | | Overlap. Coeff. |
|---|---|---|---|---|---|---|---|---|---|---|---|---|
| | | | | Mean | SE | 95% CI Lower | 95% CI Upper | Mean | SE | 95%CI Lower | 95%CI Upper | |
| Aesthetics | | | | | | | | | | | | |
| Age | 0.145 | 0.179 | 7.26*** | -0.280 | 0.012 | -0.304 | -0.256 | -0.135 | 0.016 | -0.166 | -0.104 | 0.823 |
| Attractiveness | -0.292 | 0.433 | -17.30*** | 0.400 | 0.010 | 0.380 | 0.419 | 0.107 | 0.014 | 0.081 | 0.134 | 0.701 |
| Physical build | -0.326 | 0.478 | -19.32*** | 0.375 | 0.010 | 0.355 | 0.395 | 0.049 | 0.013 | 0.022 | 0.075 | 0.673 |
| Resources | | | | | | | | | | | | |
| Education | 0.130 | 0.17 | 6.78*** | -0.514 | 0.011 | -0.536 | -0.492 | -0.384 | 0.016 | -0.415 | -0.354 | 0.804 |
| Intelligence | 0.227 | 0.317 | 13.09*** | 0.039 | 0.011 | 0.017 | 0.061 | 0.266 | 0.013 | 0.240 | 0.293 | 0.833 |
| Income | -0.030 | 0.047 | -1.86 | -1.411 | 0.009 | -1.429 | -1.393 | -1.441 | 0.013 | -1.467 | -1.415 | 0.804 |
| Personality | | | | | | | | | | | | |
| Openness | -0.131 | 0.189 | -7.65*** | 0.509 | 0.010 | 0.489 | 0.529 | 0.378 | 0.014 | 0.351 | 0.405 | 0.867 |
| Trust | 0.083 | 0.108 | 4.49*** | 0.508 | 0.012 | 0.484 | 0.532 | 0.591 | 0.014 | 0.564 | 0.619 | 0.866 |
| Emotional connection | 0.195 | 0.252 | 10.33*** | 0.374 | 0.012 | 0.351 | 0.397 | 0.569 | 0.015 | 0.540 | 0.598 | 0.756 |

Notes: $N_{males}$ = 4,375. $N_{females}$ = 2,685.

†p < 0.10

*p < 0.05

**p < 0.01

***p < 0.001. Wilcoxon rank-sum (Mann-Whitney) test were used for the sex difference in absolute importance and *t*-test for relative importance. We account for multiple comparisons of characteristics within the same group with Bonferroni adjustment (sets the significance cut-off at α/3). Setting the significance cut-off at α/9 (nine characteristics) returns the same conclusion. Effect size measures (Cliff's *delta* (non-parametric) and Cohen's *d*) are absolute value.

respondents to have a lower/higher level (all factors are not as important as the omitted factor). In addition, we also find a sex difference effect on absolute rating, i.e., female respondents gave, on average, 8.1 points higher for each of the nine ratings (total 73 points) than male respondents ($p < 0.001$; $t = 24.8$). It is also evident that for 7 out of 9 characteristics, females gave a higher rating than males. Likewise, we find large variation in the average and variance of the ratings given by respondents; such variations also seem to differ across sex and age (see F2 Fig in S1 Appendix). Additionally, our results do not change if we use the rank ordering of traits instead of the standard deviation from the average rating. Pair-wise correlations for all nine characteristics with standardized values are also provided in A2 Table in S1 Appendix.

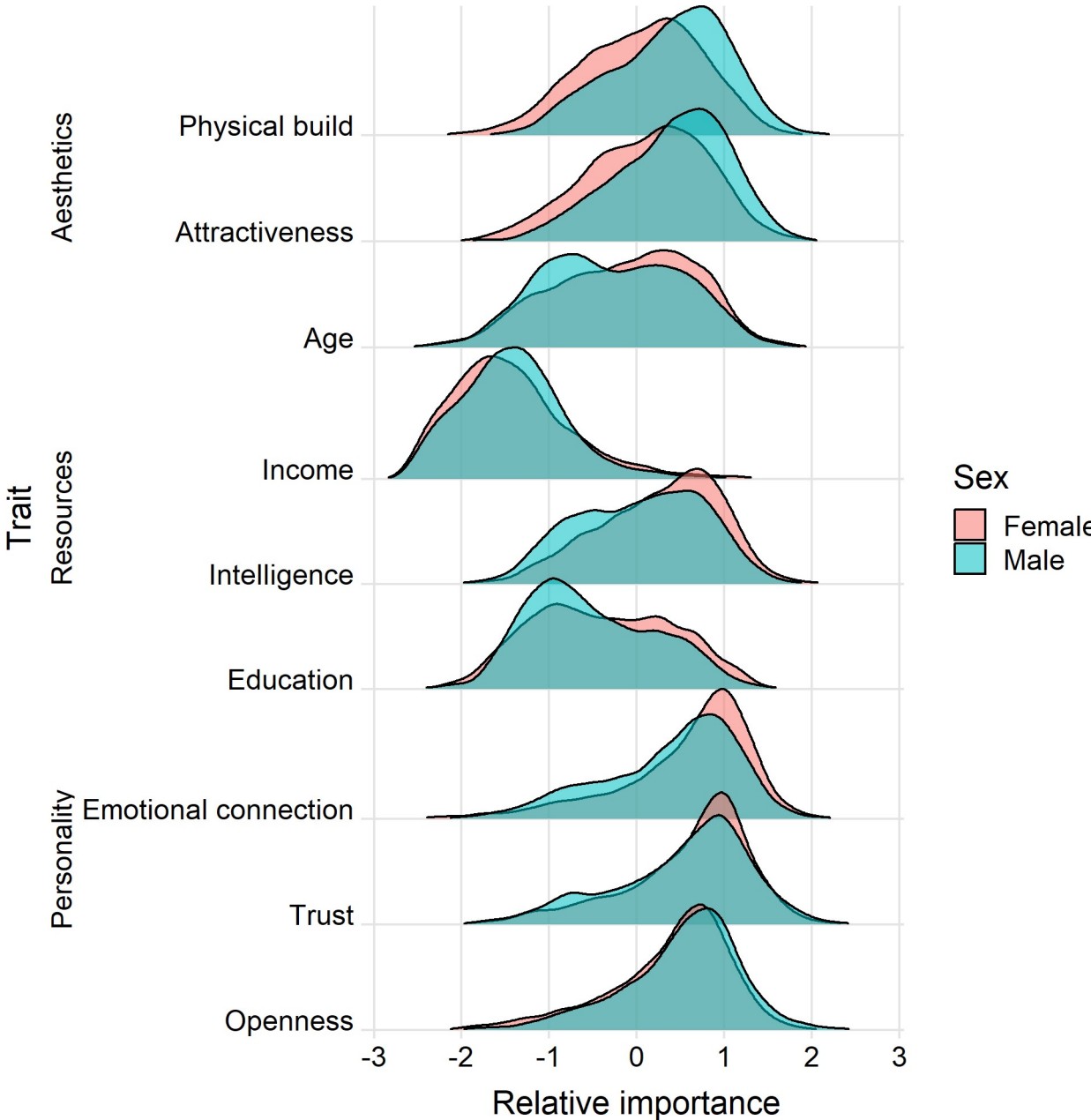

**Fig 1. Relative importance of aesthetics, resources, and personality factors for sexual attraction, by sex.** Distribution density for each factor is estimated within sex.

Compared to the distributions of the raw importance rating, we identify subtle differences in terms of sex difference of the relative importance of traits to sexual attraction (Fig 1). First, we find that, in terms of aesthetic factors, males regard both *attractiveness* (Cohen's $d$ = -0.433, $p<0.001$) and *physical build* ($d$ = -0.478, $p<0.001$) as more important characteristics for sexual attraction, relative to all other traits, compared to females, while the latter regard *age* ($d$ = 0.179, $p<0.001$) as relatively more important. The sex differences in *attractiveness* and *physical build* are substantial; for example, males rate the two factors .29 and .33 SD higher than the mean ratings given for all nine traits, whereas females rate them .11 and .05 SD higher than

their average rating, respectively (see *t*-test results in Table 1). These two factors also have the lowest overlapping coefficient, signifying the magnitude of the sex difference. Second, while both sexes regard *income* as the *least* important factor for sexual attraction, after adjusting for the variance in individual ratings, we do not find a significant difference between males and females ($d = -0.047$, $p = 0.190$). Compared to males, females place the other two resource factors, namely *education* ($d = 0.130$, $p<0.001$) and *intelligence* ($d = 0.227$, $p<0.001$), as relatively more important. Third, we find that despite females giving a higher absolute rating to *openness* compared to males, if compared with the importance of other traits, males actually regard *openness* as a slightly more important factor than do females ($d = -0.131$, $p<0.001$).

Next, we examine whether the expressed preferences for the nine characteristics, as well as their respective sex differences, covary with age. We first present graphical evidence in Fig 2, which indicates the average relative importance of the nine characteristics (standardized within respondents) across sex and age. To show potential non-linear (e.g., curvilinear) relationships with age, we use a local cubic polynomial smoothing on the average importance. For transparency, a linear fit and the raw difference between sexes are also plotted. We also show the sex differences across age in A3 Fig in S1 Appendix. Similar to the previous findings, we find that males exhibit stronger preferences for *attractiveness* and *physical build* (relative to other traits) across all ages but weaker preference for *age*, compared to females. We find that, while the relative importance for *age* and *attractiveness* decreases over age for both sexes, the preference for *physical build* increases over age for females and remains flat for males over age. There is also a tendency that sex difference in preferences for *attractiveness* and *physical build* decreases over age (A3 Fig in S1 Appendix). With respect to resource factors, we find that both sexes seem to regard *education* as relatively less important on average (females' preference is slightly stronger than male except at late 20s and early 30s), and exhibits a decreasing trend over age. It should be noted that (on average) there is an increase in female preference for *education* in the age 60+ group, despite the small number of observations. We also see a decrease in importance of *intelligence* for both sexes. However, females' preference for *intelligence* is stronger than that of males, and this difference seems particularly strong in the mid-20s and late 40s. Again, we do not see any significant sex difference in terms of *income* as both males and females regard it as the least important factor; however, one should note that (on average) younger people regard *income* as less important than older people. Lastly, we find that the preference for *openness* and *trust* increases over age for both sexes. Across all ages, females deem *trust* as relatively more important compared to males. This sex difference in *trust* appears to decrease with age, while older males regard *openness* as a relatively more important factor than do older females. We find that the relative importance of *emotional connection* for both sexes remain at the same level across age, while noting a small positive deviation for females in the early 30s and late 50s.

## Multiple regression analysis

As our study seeks to descriptively explore sex differences in perceived importance of general *aesthetic*, *resource*, and *personality* factors in relation to sexual attraction, we first perform a principal components analysis on the nine characteristics. Our results (A3 Table in S1 Appendix) show that the nine characteristics fit well into the three principal factors with eigenvalue larger than 1 (cumulative proportion of variance explained = .64). In particular, each characteristic is shown to have high (at least 0.5) and positive factor loadings on the principal factors identified. Utilizing these three factors (*aesthetic*, *resource*, and *personality*), in conjunction with the original nine characteristics, we conduct a series of regression analyses to explore factors influencing the level of importance that males and females place on each characteristic.

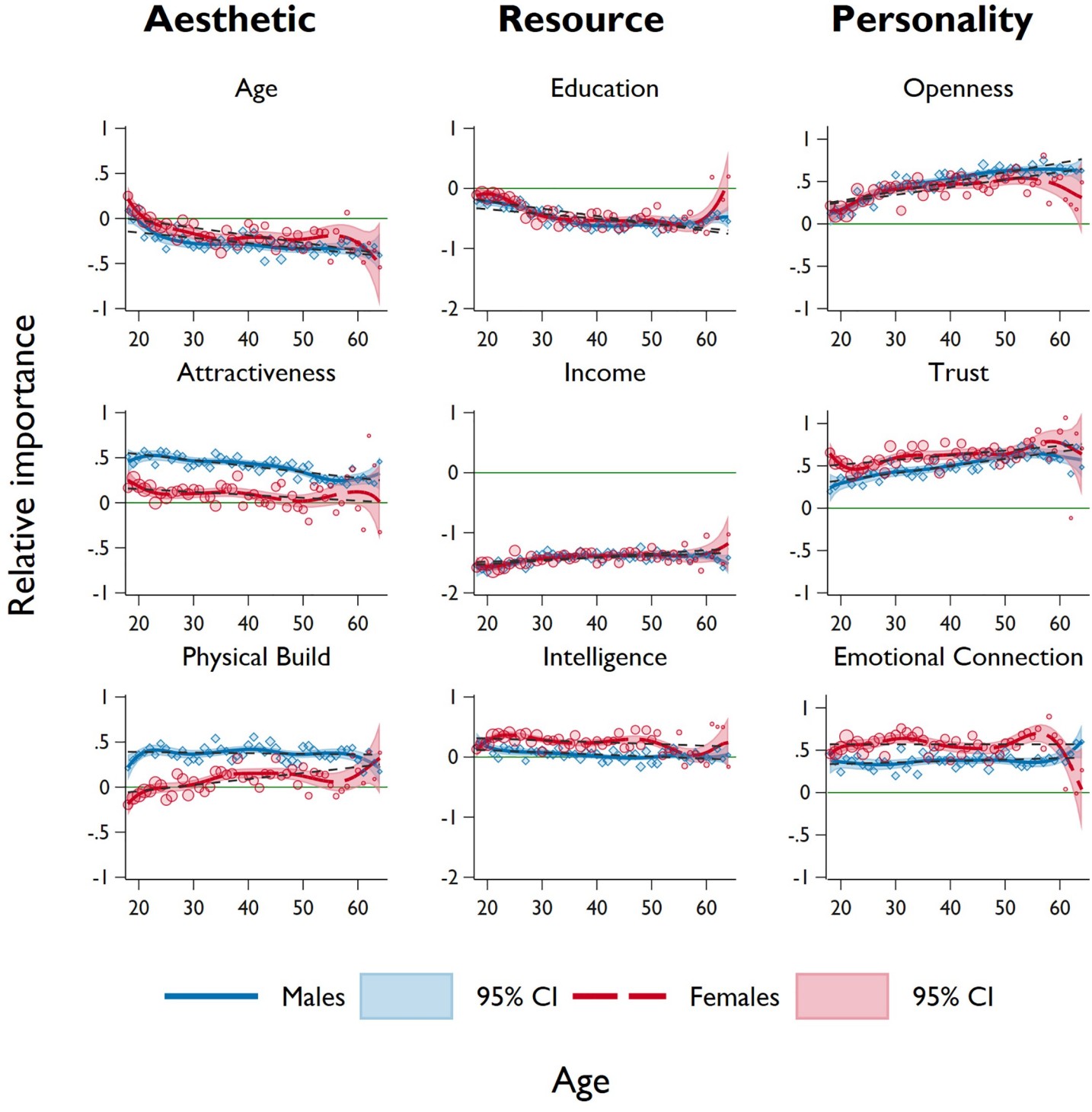

**Fig 2. Level of sexual attraction for aesthetic, resource, and personality characteristics over age, by sex.** Markers represent the average relative importance for each characteristic within sex (males = blue, females = red), calculated at each year of age. Smoothed lines represent the local cubic polynomial with Gaussian kernel function and bandwidth of 5, with shaded areas representing the 95% confidence intervals. We also show the linear fit of the observations across age, represented in grey dashed lines. The green reference line (relative importance = 0) indicates a factor is of the same importance to the average of all nine characteristics. Each characteristic is graphed for participants aged 18–64 years.

We perform the same within-subject standardization on the three factors as implemented on the nine characteristics. For the majority of characteristics (with the exception being *Resources*), we find that sex differences are at most quartic with respect to age. As the results still capture the lower-order effect (taking into account the higher-order age effect), we present the results of the cubic.

We first estimate the following model to see how sex difference in mating preferences change across age, by controlling for other factors:

$$Y_{ik} = \beta_0 + \beta_1 male_i + \beta_2 age_i + \beta_3 age_i^2 + \beta_4 age_i^3 +$$
$$\beta_5 male_i * age_i + \beta_6 male_i * age_i^2 +$$
$$\beta_6 male_i * age_i^3 + \gamma \mathbf{X}_i + \epsilon_i, \tag{1}$$

where $Y_i$ is the relative importance of characteristic *j* to respondent *i*, and $\mathbf{X}$ is the vector of control variables such as physique, education, income, marital status, sexual orientation, and self-rated happiness, health, and attractiveness. Because of the inherent sex difference bias in the distribution of variables such as height, income, or self-rated health, we standardize the controls within sex. We capture both the linear and non-linear age effects (i.e., if the sex difference changes across life span) by interacting the male dummy variable with age and its squared and cubed term. For each regression conducted, we employ an ordinary least squares (OLS) regression model with heteroscedasticity-consistent standard errors.

By holding other factors constant, we visualize the sex difference on the relative importance of each characteristic (compared to how they perceived the importance of the other factors) across age in Fig 3. The full regression results are provided in A5 Table in S1 Appendix. For most characteristics (the exception being *Resources*), we find that sex differences are at most quartic with respect to age. Since the results would still capture the lower-order effect (while taking into account the higher-order age effect), we present the results of the cubic relationship in the main text. For transparency and to assist interpretation of the results, whe also report the regression results using quadratic and linear age effect in A5 and A6 Tables in S1 Appendix. In terms of *aesthetic* (Fig 3A), we find that the consistent and significant positive sex difference decreases relatively *linearly* across age. We see a small negative sex difference in the preference for *age*, but this does not appear to differ across age. In terms of *physical build*, we also observe that the positive sex difference is strongest in younger people (under 30s) while relative preference for *attractiveness* does not seem to change across age (Fig 3B). None of the coefficients of the interaction of sex and linear, quartic, and cubic age terms are significant.

Next, we observe that females place *resource* as a more important factor than do males overall (compared to *aesthetic* and *personality*), but such difference appears to increase with age till age 30 and decrease beyond (Fig 3C). It seems that sex difference again increases after age 50, however, there is not enough evidence to provide a definite conclusion (large confidence intervals). There are no sex differences across all ages in terms of preference for *income*, confirming our earlier observations (Fig 3D). Females in their late 20s and early 30s regard *education* as more important (relative to how they perceive other characteristics) compared to their male counterparts, while females of other ages place a higher relative importance on *education*. However, we observe a strong U-shape relationship with age for sex difference in the preference for *intelligence*, showing that females in their early 40s regard *intelligence* as far more important than other factors compared to males of the same age bracket.

Lastly, we observe that sex difference in preference for *personality* (where females care relatively more than males) is largest for younger respondents. This difference appears to decrease with age before 30 and remain stable until around 55, where the sex difference is then no longer significant. The two slopes on *openness* and *trust* exhibit a positive gradient with respect to

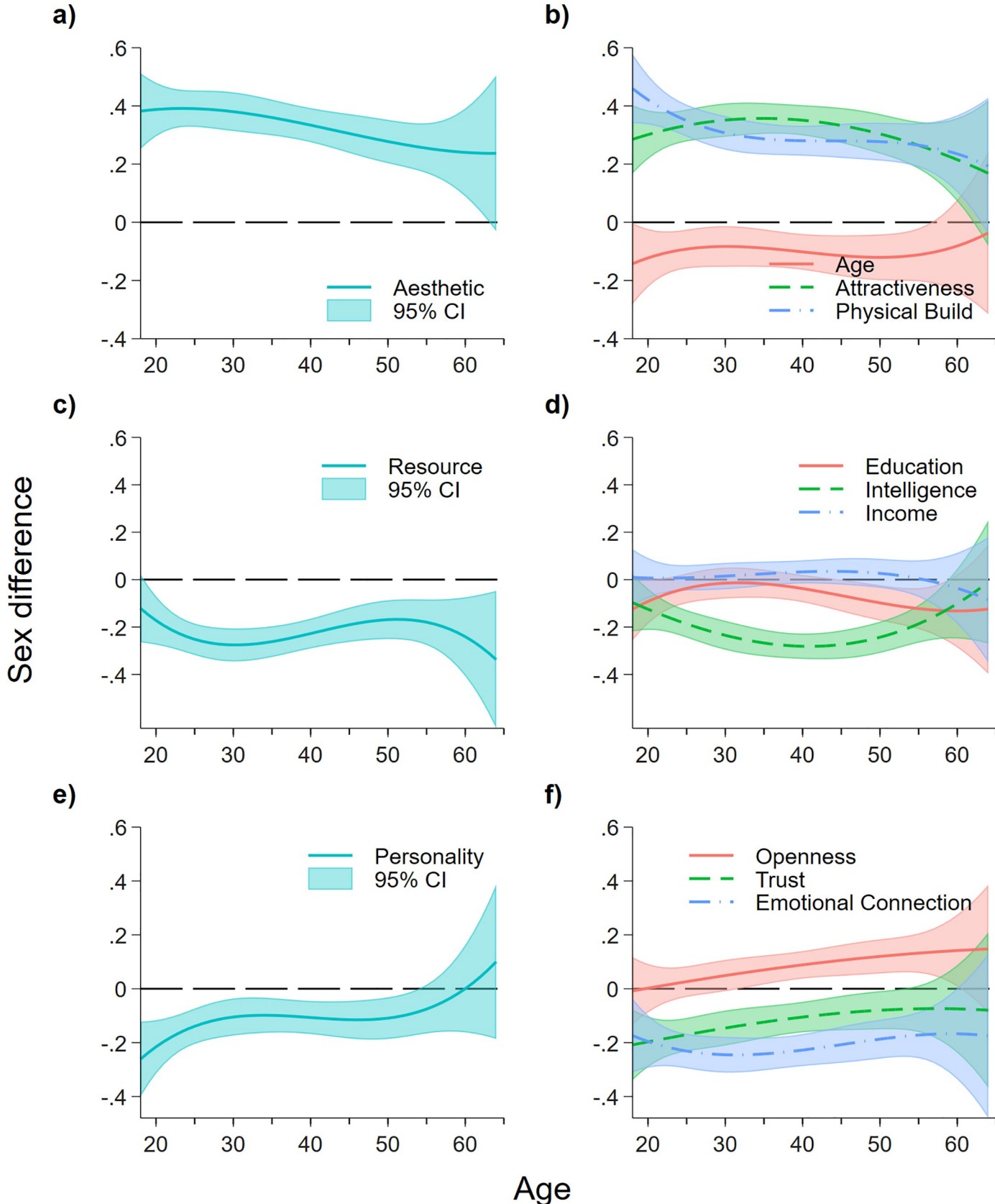

**Fig 3. Sex differences in importance of aesthetic, resource, and personality factors for sexual attraction, across lifespan.** Positive (negative) sex difference shows males (females) place higher relative importance in terms of sexual attraction to a characteristic; sex difference not significantly differ from 0 means the characteristics have equal importance to both males and females, relative to other factors. Estimates are obtained from Eq (1) using OLS. Shaded areas represent 95% confidence intervals.

age. This aligns with our earlier observations: for males, the increase in importance of these two factors across age is higher than that of females, leading to larger positive and smaller negative sex difference for *openness* and *trust*, respectively. With respect to importance in *emotional connection*, the difference in sex is significant across age, but we also note that the gap increases from age 18 to 30 and decreases thereafter.

We further explore the relative differences observed across age in Fig 3 by considering possible non-linearity of sex differences in attractiveness across participants' age, mediated by the relative importance of age as an attractiveness characteristic (see Fig 4A (without controls), and 4b (with controls)). The sex difference is more dominant (darker coloration) among those who care about age (positive value on the y-axis). We also see that sex differences in the relative importance of attractiveness are largest for the age group around 30 to 40. Those sex differences are driven by the male population aged 30 to 40 who care the most about age, as evidenced in A4 Fig in S1 Appendix.

In Fig 4C we show that sex differences in relative importance of intelligence are greatest for the age group 40 to 55 years and those who have a higher preference for the importance of attractiveness. A5 Fig in S1 Appendix indicates that for low values of importance of attractiveness there are almost no sex differences across age. Among the higher values, on the other hand, the sex difference was driven by males caring relatively less about intelligence (compared to females), in particular around the age range 40 to 55. When looking at the importance of age rather than attractiveness as a mediator of age (Fig 4D) we see an age shift of sex differences to the left, which means that the strongest differences are found in the cohort between 35 and 45 among those who place high relative importance of age. Those differences are driven by (younger) females caring relatively more about intelligence than do the male cohort of the same age (see A6 Fig in S1 Appendix). Comparing A5d and A6d Figs in S1 Appendix indicates that older females who care more about age as an attractiveness factor care less about intelligence, while the male pattern seems to be quite similar for both the importance of attractiveness and age. Thus, the shift of the sex differences between Fig 4C and 4D are driven by females.

In addition to respondent's age, we also examine how other variables such as physique, education, income, marital status, sexual orientation, and self-rated happiness, health, and attractiveness influence the perception of sexual attraction. To model whether the effects of these variables are sex-specific (e.g., present in one sex but not the other) and whether the effects differ by sex we include interaction terms between sex and these variables. For transparency, we also present regression results with male and female subsamples in A7-A9 Tables in S1 Appendix (with quartic age effects), as one can assess the raw effects of each variable on the sexual preferences for the two sexes. Since we focus on each subsample, we use the original (non-standardized) variables and present the beta coefficients (in italics) to assess the effect size in terms of unit and standard deviation change in the independent variables. For comparison of the effect size between each variable across sexes, we again employ the variables standardized within the two sexes.

We summarize the regression results on the three principal components (*aesthetic*, *resource*, and *personality*) in Fig 5 and present the results on the nine individual characteristics in A7-A9 Figs in S1 Appendix. Sex-specific age effects are included in the model, but we omit the results on age as they do not differ from the earlier regression results. We observe that those

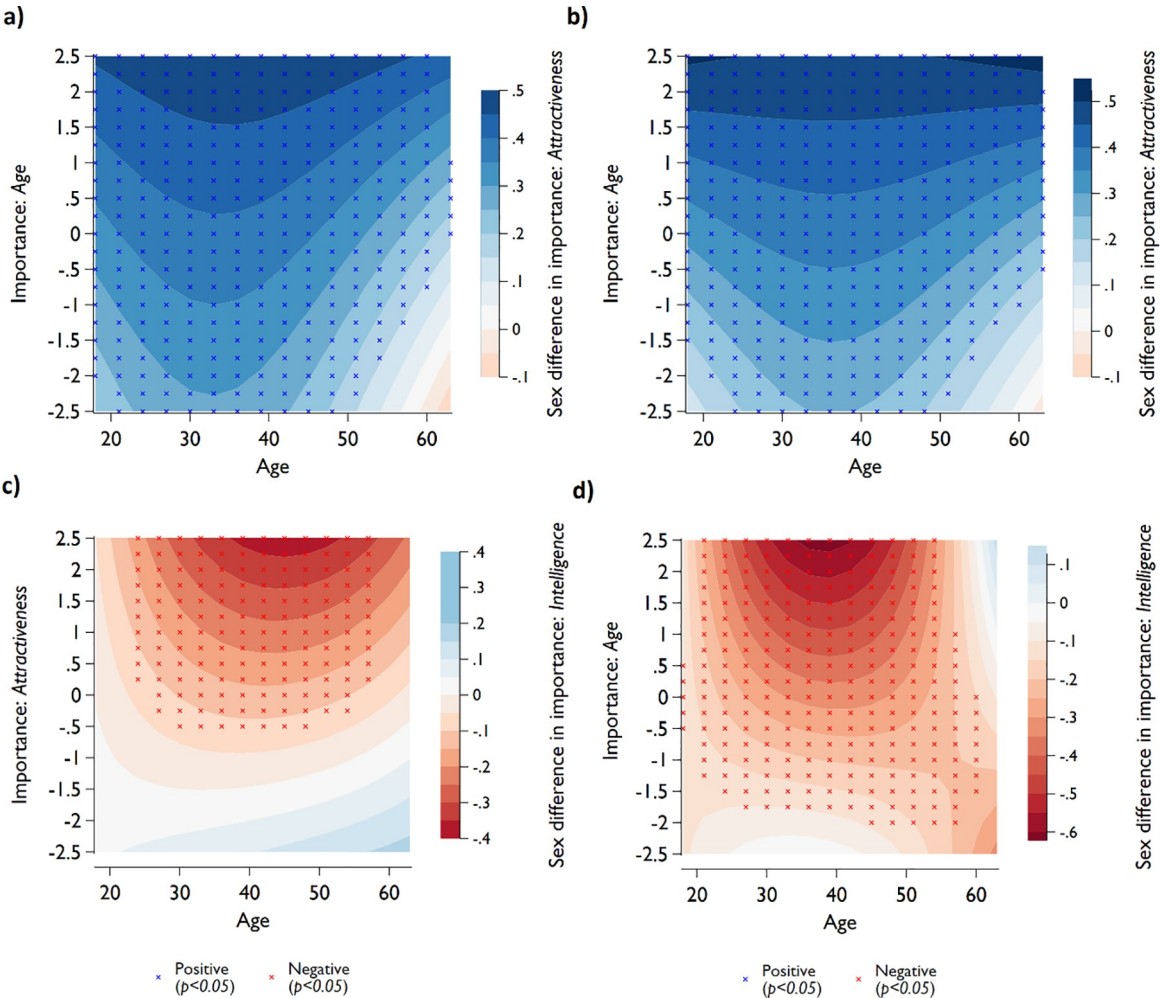

**Fig 4.** Sex differences in importance of *attractiveness* for sexual attraction across participants' age, mediated by relative importance of *age* for sexual attraction (panel **a** and **b**) and sex differences in importance of *intelligence* across age, mediated by relative importance of *attractiveness* (panel **c**) and *age* (panel **d**). Estimates of sex differences (represented by color) were obtained from OLS regressions. In each regression, we include the interaction term between sex, importance of age or attractiveness (y-axis) and age and age square (x-axis). Positive sex differences (blue) indicate males place relatively higher importance on the explained trait (z-axis) compared to females, relative to other factors. In panel **a**, no control variables were added to the regression, in panel **b**, **c**, and **d**, control variables were included in the regression model.

with higher education place less emphasis on *aesthetics*, with males even less so compared to females. Those who are single and heterosexual place higher emphasis on *resources*, and again, males show a greater preference when compared to females. Interestingly, the opposite is true for personality, with more attractive males and single males caring less about personality than their female counterparts. Further, males with offspring care less about *personality*, but females with offspring care more about *personality*.

## Discussion

Mating market preferences and decisions regarding attractiveness are arguably based on three core areas: appearances (aesthetics), personal characteristics and qualities (personality), and the ability to provide (resource) access and security to potential suitors. As our study shows, individual differences between preferences for each of these characteristics differ between

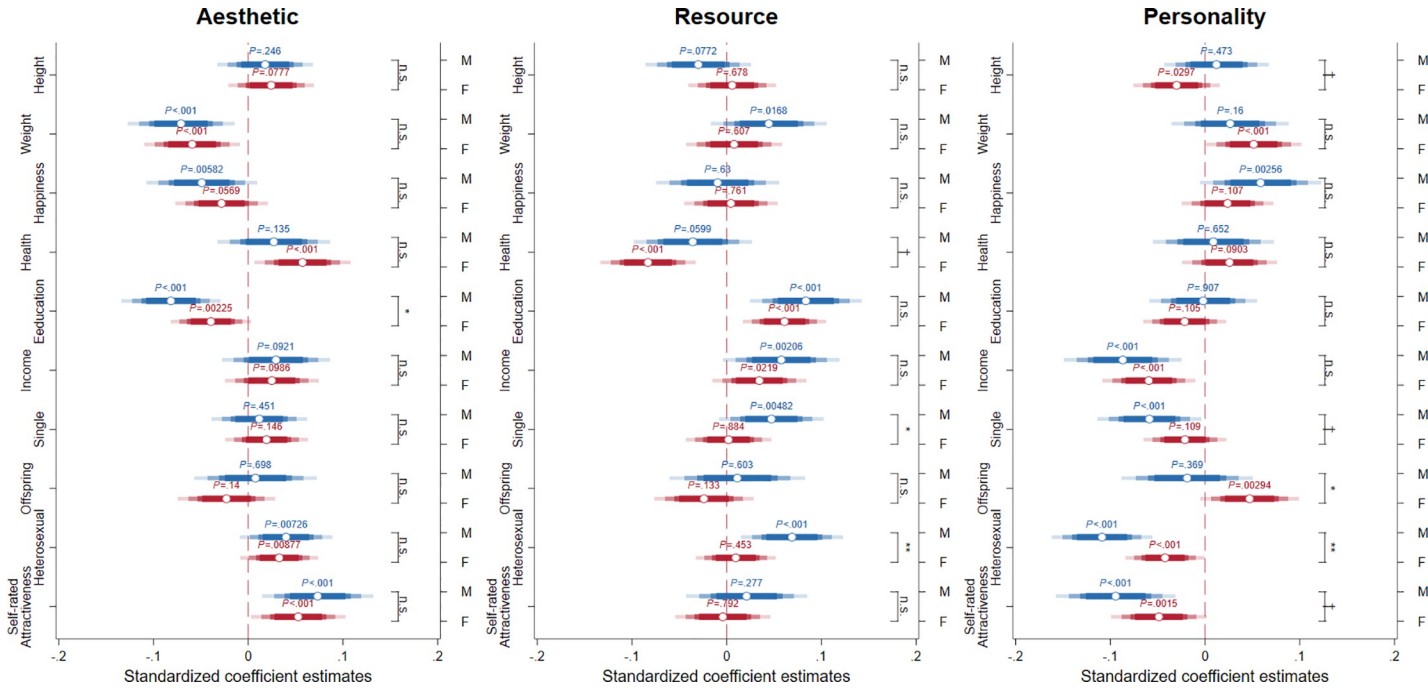

**Fig 5. Sex-specific factors on sexual attraction importance of aesthetic, resource, and personality factors.** We show the effect of each independent variable on the relative importance of characteristics of interest for both sexes (effects on male and female sexual preferences are indicated by blue and red markers, respectively). P-values of the sex-specific effects are shown above the corresponding markers, with error bars indicating 90%, 95%, 99%, and 99.5% confidence intervals. The statistical significance of the sex difference of each independent variable (i.e., interaction terms with sex) is shown to the right of the coefficient estimates. † $p < .10$; * $p < .05$; ** $p < .01$; *** $p < .001$.

women and men, as well as with age. Despite significant sex differences, however, men and women gave broadly similar priority to the measured preferences, consistent with a model of mutual mate choice [6] or the broader gender similarities hypothesis [5].

At its simplest, our study's descriptive findings demonstrate that for all nine characteristics of interests, both males and females show similar distribution patterns in their preference responses. That said, there are statistically significant sex differences within traits for eight out of the nine traits explored; on average, females rated age, education, intelligence, income, trust, and emotional connection around 9 to 14 points higher than males on our 0–100 scale range. On the surface, one may make the observation that for the population sampled, and compared with males, females care more about a greater number of characteristics when considering attractiveness in a potential mate. Such findings lend confirmatory weight to previous research findings and broader historical evolutionary theory that predicts that females tend to be choosier than men [11, 12]

By standardizing the responses to the nine traits within subject, our relative importance analysis forced an effective ranking of the nine measured preferences. Interestingly, our findings indicate greater male priority for *attractiveness* and *physical build*, compared to females, relative to all other traits. For example, males rated *attractiveness* .29 SD and *physical build* .33 SD higher than the mean ratings (to all nine traits) given; whereas females rate *attractiveness* and *physical build* .11 SD and .05 SD higher than their average rating, respectively. Conversely, compared to males, females place relatively more importance on the two resource factors, namely *education* and *intelligence*. Such results are in line with previous research findings supporting sex differences according to the predictions from parental investment theory [1, 12]. Forced ranking of preferences exposes small but detectable differences in relative emphasis on

preferences that are consistent with male resource-holding and female fecundity-nubility being important considerations in mate choice [48–50].

Our study also explored variation in perceived importance for sexual attraction of the nine characteristics, as well as their respective sex differences at different life stages. Our most novel findings again center on *attractiveness* and *physical build* (relative to other traits), with males exhibiting stronger preferences (than females) for both, across all ages. Interestingly, for both sexes, preference for attractiveness appears negatively correlated with age, but preference for *openness* and *trust* is positively associated with age. In many mating preference studies, the focus is on young adults, which means that we know relatively little about older cohorts' preferences. The consonant changes shown by women and men with age suggest one possible source of age-dependent assortative mating, consistent with predictions that mutual mate choice may be worth consideration in addition to sex-dependent preferences [6]. Age-assortative preferences warrant further research.

The study also explored non-linearity in sex-difference preferences for intelligence and attractiveness across age, mediated by the importance of age: when exploring intelligence, we checked attractiveness as a mediator. Sex differences across age are the smallest for those who reported the lowest preferences for aesthetics (age and attractiveness); however, for those who care more about aesthetics, there is a larger sex difference and such differences depend on participants' age. The sex differences in the preference for attractiveness were driven by the male cohort who cared more about age aesthetics, and were largest for the age group 30 to 40. Sex differences in the importance of intelligence were also positively affected by the importance of attractiveness and age, but sex differences for those with high aesthetic preferences were driven by females caring relatively more about intelligence, particularly for females age 40 to 55. Such findings indicating distinct variation within sex at key life stages may again speak to theories of sexual selection pressures resulting in biologically specific adaptions [11, 12].

Our multiple regression analysis explores factors impacting preferences for all nine characteristics individually, as well as their three groupings. Here, we find a consistent statistical sex difference (males relative to females) that decreases linearly with age for *aesthetics*. The opposite is true for *resources* and *personality*, with females exhibiting a stronger relative preference, particularly in the younger cohort of our sample.

Finally, our principal component regression results demonstrate interesting associations between individual differences in personality traits and our measures of preference, indicating a clear relative sex differences for single males' preferences for resources compared to females. More highly educated females express a higher relative preference for *aesthetics*, and more attractive females exhibit a higher relative preference for *personality*. We also find absolute differences for females with offspring, who place more emphasis on *personality*, whereas males with offspring report this trait as less important.

Overall, our study provides descriptive findings concerning sex and individual differences in self-reported mating preferences, most of which are consistent with predictions made by existing theories about attraction to *aesthetic*, *resource*, and *personality* traits. That so many of our findings align with theories of both parental investment and mutual mate choice speaks to the fact that the broader discipline of evolutionary mate choice research in humans still contains considerable scope for further inquiry before reaching any unified theory. The fact that such rapid advances in modern technology (such as the internet, and big data more broadly) now allows behavioral science a gamut of new avenues for analysis suggests a growing opportunity for more rigorous analysis and continued scientific debate on the topic of human mating behavior [43].

The authors acknowledge several limitations to the current study. Firstly, our sample population is the result of self-selection; naturally, any online open access national survey generates an unavoidable selection bias. While our sample population is extremely large compared to previous mate choice studies ($n = 7325$), it is important to acknowledge limitations due to

representativeness of the Australian general population. The second problem lies with the subjectivity of the participants' ratings and self-ratings; for example, the term "sexual attractiveness" may not be homogenous in meaning or interpretation for all participants in our sample, a methodological issue that is, however, present across all fields of behavioral science research. Likewise, surveying such a large number of individuals may induce "noise" around individual decisions and responses compared to the results from a more controlled laboratory experiment setting. Nevertheless, not only were the survey questions standardized for all participants in terms of both the dependent variables and their relation to the respondent's own sexual attraction, but the study delineated nine different characteristics for which the participants made their own independent assessments. Further, the large sample ($n = 7325$) and age distribution (18–65 years) of real-world online dating participants provides a unique robustness check for comparative mate choice research that has traditionally sampled more homogenous undergraduate student samples. Admittedly, however, in 21$^{st}$ century cyber mating markets (just as all historical mate choice settings) stated preferences are not always definitive indicators of actual behavior [51]. Future revealed preference research would do well to collect longitudinal data that explored individuals' stated preference and actual mate choice decisions across time. Further, it is important to note that linear high/low scales may not necessarily be the most efficient way to capture data on preference, mainly due to participant indifference. Positive-negative scales do not necessarily allow an individual to respond with indifference, and rather only permit choice of a middle 50-point marker on a 0–100 scale. Such methodological constraints are an important and ongoing consideration for future work in this space. Finally, while the current study analyses and reports the sexual attraction preference for an extremely large population of Australian online dating participants ($n = 7325$), the authors caution over-emphasis of statistically significant results stemming from such a large sample size. Any and all descriptive analysis in the current study were reported so as to provide scientific transparency, and in accordance with the current standards across the evolutionary behavioral sciences.

At different life stages both sexes prioritize (or favor) different (or similar) characteristics in a mate. For example, given that peak female fertility is essentially restricted to the (late) second and third decades of life, it seems logical that preferences will differ between males and females across these years. But this is not to say that these differences are absolute, with parental investment being a good example; not least because modern developed societies exhibit probably the most homogenous gender roles in human history. Traits and proxies for parental care and investment are thus highly valued in both sexes–although, as our research repeatedly shows– they can differ relatively at different life stages. As such, future mate choice research would do well to take into account both relative and absolute perspectives when conducting sex difference research. Given the importance of sexual attraction in reproductive decision making, ongoing research is warranted into this large-scale decision process. That the broader field of evolutionary mate choice is yet to reach a unified theory of sex differentiated stated preference across the life span speaks to the need for greater descriptive analysis of large-scale real-world mating market participants such as those included in the current study.

## Supporting information

**S1 Appendix.**
(DOCX)

## Author Contributions

**Conceptualization:** Stephen Whyte, Robert C. Brooks, Ho Fai Chan, Benno Torgler.

**Data curation:** Stephen Whyte, Robert C. Brooks, Ho Fai Chan, Benno Torgler.

**Formal analysis:** Stephen Whyte, Robert C. Brooks, Ho Fai Chan, Benno Torgler.

**Writing – original draft:** Stephen Whyte, Robert C. Brooks, Ho Fai Chan, Benno Torgler.

**Writing – review & editing:** Stephen Whyte, Robert C. Brooks, Ho Fai Chan, Benno Torgler.

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
