## [Decision Letter · Decision Letter 0]

21 Jan 2020

PONE-D-19-32550

The importance of sexiness: The impact of biological and socio-economic characteristics on human sexual attraction

PLOS ONE

Dear Dr Whyte,

Thank you for submitting your research to PLOS ONE. I have now received the opinion of two expert reviewers in the field who have considered the manuscript. Given the differences in their judgments I had hoped to secure another reviewer, but given that others had not responded I opted to provide a review for the manuscript myself. After doing this, I felt a major revision is needed, which is likely to involve significant edits and re-analysis.

As you will see, the reviewers felt that while the manuscript could offer a lot to the field - particularly because of its large and diverse sample size - there were two issues that stood out and prevented them from recommending minor revisions. One of these issues concerned the scope of the background literature and how the hypotheses were framed and the data interpreted. The second issue was with the statistical analysis of data, and the the claims that were made about these analyses that were not supported. On these issues I agree with the reviewers.

Reviewer 1 points out that the claims of an interaction analysis are not supported, as at no point is an interaction analysis clearly tested. Reviewer 2 points out the exaggeration of certain aspects of theory and glossing over of other aspects (such as strong evidence humans exhibit mutual mate choice, as opposed to great asymmetries). Addressing these kinds of points are vital for a successful revision.

I have provided my own review below this letter and hope you will find it useful. I am sorry to bear this news, and realise these edits are asking for a very different manuscript. But without significant and major revisions to the analysis and presentation of the work, the utility of a diverse and large sample of data is lost.

We would appreciate receiving your revised manuscript by Mar 05 2020 11:59PM. To enhance the reproducibility of your results, we recommend that if applicable you deposit your laboratory protocols in protocols.io, where a protocol can be assigned its own identifier (DOI) such that it can be cited independently in the future. For instructions see: http://journals.plos.org/plosone/s/submission-guidelines#loc-laboratory-protocols

We look forward to receiving your revised manuscript.

Kind regards,

Alex Jones

Academic Editor

PLOS ONE

Additional Editor Comments (if provided):

My own concerns are as follows:

The claim that females remain unaffected by mating temporality is overstated. Females tend to prefer more masculine men facially for short term relationships (Jones et al, 2019; Psychological Science), and women seem to choose more muscular men for short term mating (Frederick & Haselton, 2007; Personality and Social Psychology Bulletin). Reviewer 2 points out more of these kinds of overstatements, and this needs to be drastically changed.

There seem to be a lot of confusion around statistical terms and practice. For example, on page 12, line 7, you describe a set of multivariate analyses. Multivariate analysis involves a set of multiple Y and single-to-multiple X (in the context of regression) variables; whereas the data described here seems to simply reflect a large number of models, split by sex of participant, for age, attractiveness, build, etc (and the other kinds of ‘grouped’ variables). This not a multivariate analysis - though you certainly have scope to do so - and so should not be described as such. The notes on the tables also claim these regressions are ‘robust OLS regressions’. It is not mentioned anywhere that these regressions are robust regression in the statistical sense of the word - indeed, robust regression penalises outliers through various methods and is therefore not ordinary least squares. Reviewer 2 also asks whether any kind of multiple correction was done, and this should be reported if done - or carried out if so. Finally, the use of the Epanechnikov kernel function as a way to test relative preferences of a given trait seems too complex for what can be approximated with linear models - if there is a specific reason for using this method it should be stated and expanded upon in the methods section before it is introduced. The difference between female and male groupings at different age categories was also confusing.

In addition to addressing the queries of both reviewers, here is an analytical strategy that I think would make sense. The authors are free to reject this idea, but it is important to have a clear and concise analysis that allows readers to interpret what is going on in the data entirely, and without the overwhelming set of numbers in the manuscript as-is.

First, I suggest computing a principal components analysis or factor analysis of the ratings of the nine traits for the entire dataset (grouped here by Aesthetic, Resource and Personality). The labels assigned to these are fair groupings, but it is very unlikely that they will not be correlated in some fashion with one another - a simple idea would be age and income, or openness and intelligence, for example). A rigorous PCA or FA would boil this multidimensional data-space into something more coherent. You can then in a more data driven way provide a set of groupings by checking how each of the individual traits correlate with the new factors. An ideal setting perhaps would be finding 3 factors, wherein age, attractiveness, and physical build correlate strongly with the first factor, education, income, and intelligence with the second but not the first and so on - this may or may not occur, but the correlations are important.

Second, it is then possible to take the characteristics of your participants as you did in the later analyses (reported in the tables) and, in three models if you prefer, regress these demographics against the factors.

If you find, say, a factor that captures attractiveness and build, perhaps you will find that the sex of the participant (now entered as a predictor) will associate more strongly with this factor - i.e. men care more about this constellation of traits represented by the factor than women do. You can also test your interactions properly here in a correctly specified linear model, by allowing say age and sex to interact - do older men care less about this factor than women? This can be properly tested as at the moment the number of models is too much and they are isolated.

Journal Requirements:

2. Please modify the title to ensure that it is meeting PLOS’ guidelines (https://journals.plos.org/plosone/s/submission-guidelines#loc-title). In particular, the title should be "specific, descriptive, concise, and comprehensible to readers outside the field" and in this case it is not informative and specific about your study's scope and methodology.

Reviewers' comments:

Reviewer's Responses to Questions

**Comments to the Author**

1. Is the manuscript technically sound, and do the data support the conclusions?

Reviewer #1: No

Reviewer #2: Yes

2. Has the statistical analysis been performed appropriately and rigorously? 

Reviewer #1: No

Reviewer #2: Yes

3. Have the authors made all data underlying the findings in their manuscript fully available?

Reviewer #1: No

Reviewer #2: Yes

4. Is the manuscript presented in an intelligible fashion and written in standard English?

Reviewer #1: Yes

Reviewer #2: Yes

5. Review Comments to the Author

Reviewer #1: Thank you for inviting me to review this paper. The study is a large sample's report on the perceptions of attractive features in others. The topic is interesting and it is good to see large sample science.

My comments by paper section

Abstract

I found the wording of the results in the introduction a little confusing. In particular, it is not made clear how an interaction analysis can reveal a sex difference effect (“our interaction analysis across age pinpoints a stronger male preference for attractiveness and physical build but a stronger female preference for intelligence, trust, and emotional connection.” This latter part appears to be the interaction: “Though of these sex differences show large variation or change at different life stages for both sexes”

Introduction

It would be good to cite more work critical of the good genes hypothesis. Research on health and behavioural outcomes of attractive physiology since Trivers (1972) and Scheib et al (1999) suggest that this is not as simple as is suggested in the opening background paragraph. Some of the key findings in good genes do not replicate (Thornhill & Gangestad, 2006; Foo, 2017). Most recently, Cai et al (https://psyarxiv.com/hnbv7/) show this with health outcomes related to attractiveness, averageness, femininity and coloration. Whilst I would not expect a comprehensive review to be added (I am aware of space limitations), it would be good to see some caveats in the largely uncritical background.

It would be useful and transparent to end the introduction with operationalised hypotheses. This would also add clarification as to why the selected ‘personality traits’ are used. (I appreciate some of this appears in the method section but given the theory-informed nature of this study hypotheses and predictions would be expected).

It would useful to have a summary of the evidence of the relationship between self-reported attractiveness preferences and tested influence of attractive features. I am not familiar with evidence that the relationship between self-reported importance of, say intelligence, for attractiveness and the contribution that intelligence has in overall perceptions of attractiveness on meeting new people. This content would strengthen the justification for the paper’s core methodology.

Method

“Each question thus asks respondents to rank the level of importance they assign to each characteristic in relation to sexual attraction on a sliding scale from 1 to 100.” -> does this mean that the concrete activity was to produce a hierarchy (ordinal) data or was each attributed presented and intended to be evaluated in isolation?

Results

Table 1 should include some effect size metrics, especially given the sample size. Perhaps even include metrics such as the overlap coefficient (see Inman and Bradley, 1989). For example, the largest difference between sexes on Physical build/features t=19.32 has an 81% overlap in distribution of responses which is informative for a reader for understanding what a difference between the sexes mean in a concrete sense. The transparency of the table could be improved by clarifying ‘upper’ and ‘lower’ headings.

The explanation of figure 2 invites the reader to draw inference from their perceived difference in the difference in effect of linear prediction across age to understand sex differences in the sample. I would strongly advise against this. This would encourage the testing of an age*sex interaction to demonstrate this effect empirically.

As far as I can tell, other than the mentions in the abstract and discussion there is no empirical evaluation of the interaction between sex and age in this study. Tables 2, 3 and 4 test the effects of age within the sexes but not across (using an interaction, Fisher’s z’ or Steiger’s T/z tests) the sexes. i.e. Tables 2, 3 and 4 column 1 conducts analysis on only the 4106 male participants and then column 2 conducts analysis on only the 2519 female participants. Therefore, there are concerns about making comments on the observed difference in the relationships displayed by the sexes. The authors refer to interaction analysis

I strongly recommend that the authors consider including sex*age interaction effects on the traits or they report tests of the size of different effects (i.e. Steiger’s T).

The authors do refer to interactions throughout but no interaction tests appear in this version of the manuscript.

Discussion

The authors conclude “More specifically, our interaction analysis of sex differences in preferences across age…” however I am not able to find any interaction tests in the current paper.

The authors note at the end of their discussion “Admittedly, however, in cyber mating market settings, stated preferences are not always a good indicator of actual behaviour”. This should have much more prominence in the discussion (and perhaps introduction) as it caveats the implications of the research significantly.

Overall this paper adds a general description of sex differences in self-reported points of attraction to the literature. There needs to be inferential tests of the sex*age interaction and the paper’s final caveat should be much more prominent as the study is reliant on the honesty of those engaging with “cyber mating markets”

Reviewer #2: This review uses a large dataset from Australia to examine sex differences in mating preferences for traits related to physical attractiveness, the ability to provide resources, and three different facets of personality important for pair bonding.

There are numerous merits to the article. It uses models which account for linear and curvilinear relationships with age, which gives an indication of lifetime trajectories of mating preferences (and their associated sex differences). This is particularly relevant given that a considerable amount of mate preference research is conducted on young adults. Should sex differences in preferences changes across the life span, then conclusions drawn from young adults alone may exaggerate or understate sex differences.

The methodology of the paper and the chosen forms of analysis are sound. And the results could make a decent theoretical contribution to the area. However, the paper has some weaknesses that make it unsuitable for publication in its current form.

The main weakness is that it presents many data, but does very little in the way of actually interpreting the results. That is, it is too descriptive, and the precise merits of looking at the changes in mate preferences across age is missing from the narrative. A second weakness is the lack of specific details surrounding the choice of the statistics and presentation methods used, which makes the benefits of this approach relative to others hard to decipher. I recommend that the article be revised and resubmitted, with substantially expanded results and discussion sections, and a focus on the novel contributions to knowledge this analysis provides.

Below, I have given some more specific recommendations that I hope may improve the paper. Those which I consider to be major issues, I have marked with an asterisks.

Abstract: I find it odd that t-tests are mentioned here rather than the OLS regression and kernel-smoothing plots, which I think give a much more nuanced interpretation of the data.

Page 3, line 102. It is worth considering that the good genes hypothesis isn’t the only explanation for the preference for physical attractiveness. In fact the sexy sons hypothesis (or should we say, sexy offspring, in paid bonded species) appears to be a stronger force for selection. See a recent meta-analysis of 55 species by Prokop et al. (2012).

*Page 4, line 117. The preceding paragraph polarises the sex differences between the sexes by applying the logic of MCFC (Males compete, females choose) species to humans which have mutual mate choice. There is a large sex differences in obligate parental investment in humans, but sex differences in *typical* parental investment is much smaller, leading to similar choosiness in the long-term mating domain of humans. Humans are a MMC (mutual mate choice) species, and sex differences are relative, rather than absolute. Furthermore, polarising sex differences can be detrimental to scientific communication and pedagogy (see Stewart-Williams and Thomas, 2013). I urge the authors to acknowledge the reduced sex differences in pair-bonded species like humans by tempering their narrative here.

Page 4, line 120-3. As above, physical attractiveness is *relatively* more important for men, but is still absolutely important for both sexes. And this effect appears to be present in different cultures, even when mate preferences are constrained: see Thomas et al., (2019) for a recent study looking at samples from the UK, Norway, Australia, Singapore, and Malaysia.

Page 4, line 133-4. What does this study add that is not already known? This question has been examined countless times since the Second World War, with an additional focus on cross-cultural differences since the late 1980’s. There is a unique contribution here that I think is important, but the authors are expecting the reader to go out of their way to figure out what it is.

Page 5, line 140. The physical health = attractiveness connection is not as clear as initially thought (for a meta-analysis see Weeden & Sabini, 2005). More likely, men’s increase preference for physical attractiveness relates to its association with youth and, therefore, fecundity.

Page 6, lines 169-170. Are these economic factors the only driving force behind these preferences? It seems odd that the previous section relies on evolutionary arguments, but this one considers the driver for resource provision as the consequence of modern day market forces – especially seeing as provision for neonatal offspring is one of the most clear-cut selection pressures faced by humans throughout our history.

Page 6, lines 179-181. Is this really the case? Thomas et al., (2019), for example, found that culturally disparate groups, with different views on sex discrimination and egalitarianism, showed remarkable consistency in their preference for one of the personality traits key in successful pair bonding: kindness.

*Page 7, line 195. The fact that relationship context is not given, and that relationship preferences vary across short- and long-term domains (see Stewart-Williams et al., 2017; Thomas and Stewart-Williams, 2018; and Thomas, 2018) is problematic. One assumes that the participants are answering in line with their sociosexuality so that those who want short-term mates answer in one context, and those who want long-term mates answer in the others. This needs to be addressed in the discussion.

Page 8, line 220. What is the reason for men to outnumber women 1.7 to 1? Seems unusual for a national survey.

Table 1. Were any corrections made for multiple comparisons?

Page 9, lines 242-7. More clarity is needed regarding the selection of the Epanechinikov kernel function.

Figure 2. The notes should include which group the red reference line refers to.

Page 12, lines 7+. The multivariate analysis section. This whole section talks about which betas are significant and in what direction, but has very little in the way of interpretation of the results. It may be that the authors prefer to do this in the discussion section, but it is absent from there as well. The three tables present many data, and a more nuanced discussion of directions of findings, strengths of findings, and within category consistency and deviation is appropriate. There is also very little discussion of the age effects, and what implications a curvilinear pattern has for how we view sex differences.

References

Prokop, Z. M., Michalczyk, Ł., Drobniak, S. M., Herdegen, M., & Radwan, J. (2012). Meta‐analysis suggests choosy females get sexy sons more than “good genes”. Evolution: International Journal of Organic Evolution, 66(9), 2665-2673.

Stewart-Williams, S., & Thomas, A. G. (2013). The ape that thought it was a peacock: Does evolutionary psychology exaggerate human sex differences?. Psychological Inquiry, 24(3), 137-168.

Stewart-Williams, S., Butler, C. A., & Thomas, A. G. (2017). Sexual history and present attractiveness: People want a mate with a bit of a past, but not too much. The Journal of Sex Research, 54(9), 1097-1105.

Thomas, A. G., & Stewart-Williams, S. (2018). Mating strategy flexibility in the laboratory: Preferences for long-and short-term mating change in response to evolutionarily relevant variables. Evolution and Human Behavior, 39(1), 82-93.

Thomas, A. G. (2018). Lowering partner standards in a short‐term mating context. In T. K. Shackelford, & V. A. Weekes‐Shackelford (Eds.), Encyclopedia of evolutionary psychological science (pp. 1– 3). Cham, Switzerland: Springer International Publishing.

Thomas, A. G., Jonason, P. K., Blackburn, J. D., Kennair, L. E. O., Lowe, R., Malouff, J., ... & Li, N. P. (2019). Mate preference priorities in the East and West: A cross‐cultural test of the mate preference priority model. Journal of personality.

Weeden, J., & Sabini, J. (2005). Physical attractiveness and health in Western societies: a review. Psychological bulletin, 131(5), 635.

6. PLOS authors have the option to publish the peer review history of their article (what does this mean?). If published, this will include your full peer review and any attached files.

Reviewer #1: No

Reviewer #2: No

---

## [Author Response · Author response to Decision Letter 0]

6 Oct 2020

PONE-D-19-32550

The importance of sexiness: The impact of biological and socio-economic characteristics on human sexual attraction

Dear Dr Jones,

We would like to sincerely thank yourself as editor and both reviewers for such a comprehensive and informative review. As you note in your review the large and diverse sample size offers the opportunity for a unique and significant contribution to the mate choice literature. That said we also acknowledge your comments requesting major revisions. 

We have addressed all of the comments and incorporated all of the requested changes and edits, by completing a major re-write of the study. We have incorporated the majority of the new literature and theory suggested by the reviewers into the re-write. We have also conducted more and completely new analysis based on the reviewers and your feedback. 

In line with both of the reviewers comments we have completely revised the entire analysis section (Reviewer 1), and slightly re-aligned the scope of the manuscript to provide a more nuanced paper on both absolute and relative sex differences across age (Reviewer 2), with rewrites to the abstract, introduction, background and discussion.

We appreciate that because of such significant changes to the manuscript the review process will again be substantial by yourself and the reviewers, and we would like to express our gratitude for the opportunity to resubmit to PLoS One.

Thank you,

Dr Stephen Whyte

---

## [Decision Letter · Decision Letter 1]

8 Dec 2020

PONE-D-19-32550R1

The importance of sexiness: Relative and absolute sex differences in human sexual attraction at different life stages

PLOS ONE

Dear Dr. Whyte,

Thank you for submitting your manuscript to PLOS ONE. After careful consideration, we feel that it has merit but does not fully meet PLOS ONE’s publication criteria as it currently stands. Therefore, we invite you to submit a revised version of the manuscript that addresses the points raised during the review process.

We look forward to receiving your revised manuscript.

Kind regards,

Alex Jones

Academic Editor

PLOS ONE

Additional Editor Comments (if provided):

Dear Dr Whyte,

Thank you for submitting your revised manuscript, and for the care you have taken in your responses to the reviewers comments as well as my own suggestions. I have now had a chance to review your manuscript and obtain further commentary from the original reviewers.

While one reviewer recommended acceptance, the other suggested another round of major revisions. My own leaning is between these two perspectives. I find the manuscript much improved, but I agree with Reviewer 1 that the theoretical contribution of the work has not been clearly highlighted, and that there are some inconsistencies throughout the paper in terms of analyses missing from the supplementary information, as well as some grammatical errors throughout. I also agree with Reviewer 2 that the interpretation of the effects, especially with such a large sample size, are often speculative. I would urge the authors to express strong caution in their interpretation of these data, where statistical significance may well be meaningless.

Finally, I would also agree with Reviewer 1 that the authors should make every effort possible to make these data openly available, as well as responding to their comments in general.

Reviewers' comments:

Reviewer's Responses to Questions

**Comments to the Author**

1. If the authors have adequately addressed your comments raised in a previous round of review and you feel that this manuscript is now acceptable for publication, you may indicate that here to bypass the “Comments to the Author” section, enter your conflict of interest statement in the “Confidential to Editor” section, and submit your "Accept" recommendation.

Reviewer #1: All comments have been addressed

Reviewer #2: (No Response)

2. Is the manuscript technically sound, and do the data support the conclusions?

Reviewer #1: Yes

Reviewer #2: Yes

3. Has the statistical analysis been performed appropriately and rigorously? 

Reviewer #1: Yes

Reviewer #2: Yes

4. Have the authors made all data underlying the findings in their manuscript fully available?

Reviewer #1: Yes

Reviewer #2: No

5. Is the manuscript presented in an intelligible fashion and written in standard English?

Reviewer #1: Yes

Reviewer #2: Yes

6. Review Comments to the Author

Reviewer #1: Dear authors,

I am pleased to re-review this paper. The revised text is a comprehensive, nuanced and complete take on the dataset. I particularly appreciate that the authors have welcomed the use of overlap coefficients and formalised tests of the interactions. I think the paper tells a very interesting story and have spent quite some time reading the results with interest. The clarity of the figures and the comprehensiveness of the tables should be applauded. This is large sample research at it's best.

Perhaps there is an overemphasis on statistical significance values. With N>7000 inference from p values by traditional liberal criteria leads to some low thresholds of notable effects (I did note the identification of significance of p<.10 at some points - which in N>7000 is a very small effect). However, the discussion and interpretation of these results are clear and I consider this report to be in line with current standards.

The authors should be proud of this work and I wish them good luck in future similar research.

Reviewer #2: I'm afraid that despite a substantial rewrite, the authors have done very little to address the concerns I raised in my original review. I still do not believe this manuscript is suitable for publication in its current form for the reasons outlined below. I still see merit in this manuscript, and think it has the potential to make a theoretical contribution but only after a substantial revision. My recommendation to the editor is that the authors revise and resubmit this article.

Major points:

#1 The theoretical contribution of this paper is still unclear. Sex differences in the variables under consideration have been studied for decades and this research is now becoming more and more nuanced. Where does this paper fit in with the current state of the literature? A large sample isn't enough. My observation is that the change in preferences across age is the novel aspect, but this is not expressed or highlighted in the manuscript at all. For example, the literature behind sex differences in personality makes no prediction for how this pattern might change with age. The narrative needs dramatic reframing and focus. This is still an overwhelmingly descriptive manuscript with very little theoretical focus or integration.

#2 The opening paragraph makes several claims about decision making in the mating domain yet is barren of literature.

#3 There are no formal predictions made in the article (or its exploratory nature is not made clear).

#4 There is no formal start to the results section

#5 The descriptives section is incredibly long. It's broken down into fine detail for no theoretical reason. What does all of this "add" from a theoretical perspective? What is the advantage of this approach over a simple table with averages, measures of spread, and effect sizes?

#6 The table (A4) which contains the coefficients for the key regression model - the model which examines changes with age, and is arguably the most important part of the article - is missing. There are other cases of lack of attention to detail throughout the article.

#7 The discussion makes no attempt to integrate the findings with the literature, it simply provides a summary of the key patterns observed. Often times, the article reads as if the examination of sex differences among these traits in and of itself is a novel contribution to the evolutionary literature. That is not the case. I suspect that this assumption may have something to do with the literature covered which is, for the most part, quite dated.

Minor points:

#8 I recommend standardized the use of effect sizes throughout (Cohen's d or r) for easier comparison between works on sex differences.

#9 The authors indicate that the data for this article is not readily accessible as per the Plos ONE guidelines. They infer that if someone wishes to access the data they need to seek permission from a third party (their institution). The authors should seek this permission and then upload the data to a repository such as OSF, rather than ask potentially dozens of researchers to embark on this battle themselves.

7. PLOS authors have the option to publish the peer review history of their article (what does this mean?). If published, this will include your full peer review and any attached files.

Reviewer #1: No

Reviewer #2: **Yes: **Dr Andrew G. Thomas

---

## [Author Response · Author response to Decision Letter 1]

30 Mar 2021

31 March 2021

2 George St,

Brisbane, Queensland

Australia

RE: PONE-D-19-32550-R2 - The importance of sexiness: Relative and absolute sex differences in sexual attraction at different life stages

Dear Dr Jones,

We again thank you and both reviewers for helpful feedback on our revised manuscript. We have now revised the manuscript in line with recommendations from the reviewers and the editor’s comments, all of which helped clarifying the important perspectives of both reviewers. Below you will find our point-by-point response to the comments. Beyond that we have decided to change the title of the manuscript to: Sex Differences in Sexual Attraction for Aesthetics, Resources and Personality Across Age. We believe that the new title is a better reflection of the content of the paper. Moreover, we would like to note that the data are now openly available on OSF (see OSF | Sex Differences in Sexual Attraction for Aesthetics, Resources and Personality Across Age). 

We hope the revision will meet your expectations and facilitate the decision to publish our paper in PLoS One.

Reviewer #1: Dear authors,

I am pleased to re-review this paper. The revised text is a comprehensive, nuanced and complete take on the dataset. I particularly appreciate that the authors have welcomed the use of overlap coefficients and formalised tests of the interactions. I think the paper tells a very interesting story and have spent quite some time reading the results with interest. The clarity of the figures and the comprehensiveness of the tables should be applauded. This is large sample research at it's best.

Perhaps there is an overemphasis on statistical significance values. With N>7000 inference from p values by traditional liberal criteria leads to some low thresholds of notable effects (I did note the identification of significance of p<.10 at some points - which in N>7000 is a very small effect). However, the discussion and interpretation of these results are clear and I consider this report to be in line with current standards.

The authors should be proud of this work and I wish them good luck in future similar research.

We thank Reviewer 1 for their helpful comments and feedback throughout the review process. While we note the reviewer’s comments that “interpretation of these results are clear and I consider this report to be in line with current standards” we also provide extensive clarification regarding any possible over-emphasis on notable effects: 

While the current study analyses and reports the sexual attraction preference for an extremely large population of Australian online dating participants (n=7325), the authors caution over-emphasis on statistically significant results stemming from such a large sample size. Any and all descriptive analysis in the current study was reported wtihin the study so as to provide scientific transparency, and in accordance with the current standards across the evolutionary behavioural sciences.

Reviewer #2: I'm afraid that despite a substantial rewrite, the authors have done very little to address the concerns I raised in my original review. I still do not believe this manuscript is suitable for publication in its current form for the reasons outlined below. I still see merit in this manuscript, and think it has the potential to make a theoretical contribution but only after a substantial revision. My recommendation to the editor is that the authors revise and resubmit this article.

Major points:

#1 The theoretical contribution of this paper is still unclear. Sex differences in the variables under consideration have been studied for decades and this research is now becoming more and more nuanced. Where does this paper fit in with the current state of the literature? A large sample isn't enough. My observation is that the change in preferences across age is the novel aspect, but this is not expressed or highlighted in the manuscript at all. For example, the literature behind sex differences in personality makes no prediction for how this pattern might change with age. The narrative needs dramatic reframing and focus. This is still an overwhelmingly descriptive manuscript with very little theoretical focus or integration.

We agree with the reviewer that the novel aspect of the studies is the exploration of how preferences change across age. We have now restructured the paper accordingly (narrative, reframing, and focus). This means that we have placed some previous results in the main text in the Appendix and we have extended the empirical approach by adding a new figure with various contour plots showing interaction effects between the importance of age for sexual attraction and age to understand sex differences in attractiveness as well as the importance of age and attractiveness across age in regards to sex differences for the importance of intelligence. Those visualizations provide additional insights on preference changes across age – beyond what was previously achieved – and offers a better understanding of non-linear relationships. We agree with the reviewer’s statement that the literature behind sex differences in personality makes no prediction for how preferences change with age. Our comparative empirical advantage is the analysis of a dataset with a larger age distribution as (most) previous studies often have an age distribution skewed towards the younger population. We therefore believe that an empirically oriented study like the one conducted here can hopefully guide future theoretical and empirical studies; taking into account that science can be seen as a constant interaction between speaking to theorists and searching for facts and moving between phases of interpretation and the more descriptive and explorative summarization. 

#2 The opening paragraph makes several claims about decision making in the mating domain yet is barren of literature.

We have re-written the introduction, re-formatted it, and cited extra literature to provide greater clarity for the reader.

#3 There are no formal predictions made in the article (or its exploratory nature is not made clear).

We have now made the contribution clearer (see previous response) and refer in more detail to the explorative and empirical orientation of the study. 

#4 There is no formal start to the results section

We have re-formatted and titled the “Results” section. 

#5 The descriptives section is incredibly long. It's broken down into fine detail for no theoretical reason. What does all of this "add" from a theoretical perspective? What is the advantage of this approach over a simple table with averages, measures of spread, and effect sizes?

The descriptive statistics section has been removed and replaced instead with a simple table of descriptive statistics as per the reviewer’s suggestion. We now also place the table in the Appendix. 

#6 The table (A4) which contains the coefficients for the key regression model - the model which examines changes with age, and is arguably the most important part of the article - is missing. There are other cases of lack of attention to detail throughout the article.

We agree with this evaluation, but we believe that Figure 3 is more appealing to the readers for understanding what is happening in the regression results. Thus, we retain the full regression results in the Appendix. We have also checked the entire manuscript for lack of attention to details including also a careful checking and correcting of grammatical, labelling, or formatting errors. 

#7 The discussion makes no attempt to integrate the findings with the literature, it simply provides a summary of the key patterns observed. Often times, the article reads as if the examination of sex differences among these traits in and of itself is a novel contribution to the evolutionary literature. That is not the case. I suspect that this assumption may have something to do with the literature covered which is, for the most part, quite dated.

We believe that the reframing towards preferences change over age allows to emphasize the innovative nature of the contribution. The discussion section not only summarises what we can learn from the empirical results but also tries to link the results back to the previous evolutionary mate choice literature and sex difference literature (e.g., mutual mate choice (see Stewart-Williams & Thomas 2013), or the gender similarities hypothesis (see Hyde 2005), and also referencing more recent studies (see, e.g., Lassek and Gaulin 2019). The revised discussion now provides more clarity on where the key findings sit within the literature and how it informs the different views in the broader evolutionary mate choice literature. 

Minor points:

#8 I recommend standardized the use of effect sizes throughout (Cohen's d or r) for easier comparison between works on sex differences.

We have now reported the Cohen’s d (relative importance) and Cliff’s delta (absolute importance; non-parametric) for the sex differences in Table 2. 

#9 The authors indicate that the data for this article is not readily accessible as per the Plos ONE guidelines. They infer that if someone wishes to access the data they need to seek permission from a third party (their institution). The authors should seek this permission and then upload the data to a repository such as OSF, rather than ask potentially dozens of researchers to embark on this battle themselves.

This is a very good point. We now provide full access to the data via OSF | Sex Differences in Sexual Attraction for Aesthetics, Resources and Personality Across Age

---

## [Editor Report · Decision Letter 2]

1 Apr 2021

Sex Differences in Sexual Attraction for Aesthetics, Resources and Personality Across Age.

PONE-D-19-32550R2

Dear Dr. Whyte,

We’re pleased to inform you that your manuscript has been judged scientifically suitable for publication and will be formally accepted for publication once it meets all outstanding technical requirements.

Kind regards,

Alex Jones

Academic Editor

PLOS ONE
---

## [Editor Report · Acceptance letter]

16 Apr 2021

PONE-D-19-32550R2 

Sex Differences in Sexual Attraction for Aesthetics, Resources and Personality Across Age 

Dear Dr. Whyte:

I'm pleased to inform you that your manuscript has been deemed suitable for publication in PLOS ONE. Congratulations! Your manuscript is now with our production department. 

Kind regards, 

on behalf of

Dr. Alex Jones 

Academic Editor

PLOS ONE